🔓 | **Open Peer Review** | Clinical Microbiology | Research Article

# Whole-genome sequence analyses reveal cryptic species-level diversity among clinical *Actinotignum* isolates misidentified by matrix-assisted laser desorption/ionization time-of-flight mass spectrometry

Dylan M. Winterflood,[1] Guillem Seguí-Crespi,[2,3] Charlotte Peeters,[4] Claudia Lang,[1] Branislav Ivan,[1] Peter M. Keller,[1] Pascal Schlaepfer,[5] Edward R. B. Moore,[2,3] Peter Vandamme,[4] Daniel Goldenberger[1]

**ABSTRACT**    *Actinotignum schaalii* is a human commensal bacterium that is well-known for causing urinary tract infections, predominantly in the elderly. However, reliable identification of *A. schaalii* in routine clinical microbiology has been challenging, mainly due to its fastidious growth and identification issues. The genus *Actinotignum* currently comprises three species with validly published names, i.e., *A. schaalii*, *A. sanguinis*, and *A. urinale*. The former two are difficult to distinguish by routine clinical microbiology laboratory protocols, including phenotypic profiling and matrix-assisted laser desorption/ionization mass spectrometry (MALDI-TOF MS). A total of 137 new *Actinotignum* isolates were collected from urine and various other human clinical samples between 2010 and 2023 and were identified by MALDI-TOF MS as *A. schaalii/A. sanguinis*. Whole-genome sequence-based analyses identified none of these isolates as *A. schaalii* or *A. sanguinis* and demonstrated that the strains were clustered into seven genomic groups. These seven genomic groups represented seven novel *Actinotignum* species, which we propose to be classified as *A. lotii* sp. nov. (*n* = 82 isolates), *A. saccati* sp. nov. (*n* = 38), *A. inguinis* sp. nov. (*n* = 8), *A. vesiculae* sp. nov. (*n* = 3), *A. stranguriae* sp. nov. (*n* = 3), *A. cystesis* sp. nov. (*n* = 2), and *A. urematis* sp. nov. (*n* = 1). Therefore, MALDI-TOF MS analysis allows reliable identification only to the genus level for *Actinotignum* species, whereas commercial identification libraries need revision and extension to improve their diagnostic capacities.

**IMPORTANCE** *Actinotignum schaalii* is a human commensal bacterium that causes urinary tract infections, as well as invasive infections, predominantly in the elderly. Yet, isolation and reliable identification of *A. schaalii* in routine clinical microbiology is challenging. Today, matrix-assisted laser desorption/ionization mass spectrometry (MALDI-TOF MS) has been recommended for the identification of *Actinotignum* bacteria but fails to discriminate between *A. schaalii* and *A. sanguinis*. The present study used whole-genome sequence analyses and demonstrated that, among 137 new human clinical *Actinotignum* isolates tentatively identified as *A. schaalii/A. sanguinis* through MALDI-TOF MS, none belonged to *A. schaalii* or *A. sanguinis*; rather, they represent seven novel *Actinotignum* species. The commercial MALDI-TOF MS identification databases need marked improvement if correct species-level identifications will be reliable. Currently, whole-genome sequence-based identification may need to be used to provide correct species identifications and to assess the genetic and clinical differences between established and newly defined *Actinotignum* species.

**Peer Reviewer** Nidhika Berry, Public Health Wales Microbiology Swansea, Swansea, United Kingdom

Address correspondence to Daniel Goldenberger, daniel.goldenberger@gmail.com.

The authors declare no conflict of interest.

10.1128/spectrum.02190-25 **1**

**KEYWORDS** *Actinotignum*, species identification, genome sequence analysis, MALDI-TOF MS, clinical strains, novel species

*A*ctinotignum (previously *Actinobaculum*) *schaalii* is a commensal bacterium of the human urogenital tract and is involved in urinary tract infections predominantly in elderly patients (1). In addition, reports have implicated *Actinotignum* bacteria in bacteremia, endocarditis, Fournier's gangrene, abscesses, cellulitis, discitis, and spondylodiscitis (2), and recent data suggest that they play an important role in the infant male urobiome (3). Two additional species, i.e., *A. sanguinis* and *A. urinale*, are less well-known and are possibly opportunistic human pathogens (4). A fourth species was isolated from a male urine sample and tentatively named "*A. timonense*," but the name was never validated (5, 6).

Primary isolation and cultivation of these facultative anaerobic bacteria are challenging, as blood-enriched media and incubation times of 48 h in 5% $CO_2$ or anaerobic atmosphere are required for optimal growth (4), whereas urine samples are routinely incubated aerobically for 24 h on selective chromogenic or MacConkey agar (4, 7). Also, *A. schaalii* is commonly mistaken for a contaminant due to its morphological resemblance to skin and mucosal commensal bacteria, and also due to the colonies being easily overgrown by other species (8). Together, these factors compromise the routine isolation of *A. schaalii* from clinical samples and impede the assessment of its importance as a human microbiome bacterium or a urinary tract pathogen.

Identification of isolates of *Actinotignum* species has long been hindered by the inability of phenotype-based characterizations to accurately differentiate *A. schaalii*, requiring comparative 16S rRNA gene sequence analyses for more reliable identification. Today, matrix-assisted laser desorption/ionization mass spectrometry (MALDI-TOF MS) enables relatively reliable and rapid identification of *A. schaalii* and has been recommended for the diagnosis of this species (4). Nevertheless, the distinction between *A. schaalii* and *A. sanguinis* is uncertain, even when high identification scores are obtained, according to the Bruker Microflex LT system.

The present study was initiated to verify the performance of MALDI-TOF MS identification for *A. schaalii* and *A. sanguinis*. To this end, we collected a total of 137 clinical isolates between 2010 and 2023 and performed MALDI-TOF MS and whole-genome sequence-based classification.

## MATERIALS AND METHODS

### Bacterial strains

*Actinotignum* isolates were collected between 2010 and 2023 from various clinical samples in the University Hospital Basel, a tertiary care hospital in Switzerland. Urine samples were cultured on Columbia sheep blood agar, incubated in 5% $CO_2$ atmosphere, and on CHROMagar Orientation medium according to standard methods. Aerobic and anaerobic cultures of different non-urine clinical specimens were performed, according to standard microbiological procedures for cultivation of facultative anaerobic bacteria, including enrichment culture in thioglycolate broth medium. Anaerobic cultures were incubated and processed in an anaerobic workstation (Whitley A 95, Don Whitley Scientific Ltd., Bingley, UK). Partial 16S rRNA gene sequencing was applied for tentative species identifications before the introduction of MALDI-TOF MS in the diagnostic laboratory in 2012 (9). Type strains of *A. schaalii* (DSM 15541[T] = CCUG 27420[T]), *A. sanguinis* (DSM 26039[T] = CCUG 64086[T]), and *A. urinale* (CCUG 46093[T]) were analyzed as controls.

### MALDI-TOF MS analyses

MALDI-TOF MS was performed routinely as described before (9), using a formic acid on-plate extraction method (10) and the Bruker Microflex LT system. Different versions

of the Bruker database and MBT Compass software (Bruker Daltonics GmbH, Bremen, Germany) were used as they were released during the study period. Regardless of the database and software version, isolates included in the present study were consistently identified as *A. schaalii* or *A. sanguinis*. For the present study, MS spectra of 102 *Actinotignum* isolates were re-analyzed with the present database (V12) and software (version 4.1.100) versions using the in-house batch-extraction tool EasyMaldi (available at https://github.com/DylanMorris483/win_batch_EasyMaldi).

## 16S rRNA gene sequence determinations and analyses

Genomic DNA was extracted from the biomass of fresh cultures, and the nearly complete 16S rRNA gene sequences were determined, as described before (11), using PCR amplification and Sanger sequencing protocols and primers described previously (12). Sequencing reactions were performed, using the Big Dye Terminator 3.1 Kit (Applied Biosystems, Carlsbad, CA, USA), following the manufacturer's instructions, and the DNA nucleotide sequences were determined, using an ABI Prism3100-Avant Genetic Analyzer (Applied Biosystems). The obtained 16S rRNA gene sequence data were edited manually and submitted to the EzBioCloud for identification (13). The sequences were generated as hybrid assemblies of Sanger and Illumina genome sequence data and were deposited at the National Center for Biotechnology Information (NCBI) using the GenBank submission portal.

## Whole-genome sequence analysis

Draft whole-genome sequences were generated, as described earlier (14), using an Illumina platform (MiSeq or NextSeq500) and NexteraXT or NexteraFlex (Illumina, San Diego, CA, USA) for library preparation. The quality of the raw paired-end reads was assessed using FastQC v0.11.9 (15). Raw sequence reads were processed to remove adapter sequences, using Trimmomatic v0.39 (16). Quality filtering was performed, using Sickle v1.33 (17), applying a Phred score cutoff of Q30 to retain high-quality reads. The high-quality paired-end reads, along with remaining single-end reads (from discarded pairs), were subjected to *de novo* genome assembly, using SPAdes v3.13.0 (18). The quality of the resulting assemblies was evaluated with Quast v5.0.2 and CheckM2 v1.1.0 (19, 20).

## Genomic taxonomic analyses

Reference genomes of *A. schaalii* CCUG 27420$^T$ (GCF_034555335.1), *A. sanguinis* CCUG 64068$^T$ (GCF_034119385.1), *A. urinale* CCUG 46093$^T$, (GCF_034119365.1), and "*A. timonense*" strain Marseille-P2803 (LT706966.1) were downloaded from the NCBI database. The Type (Strain) Genome Server (TYGS) (21) was used to identify all isolates and to calculate the degree of relatedness toward the nearest neighbor species. Digital DNA–DNA hybridization (dDDH) values and confidence intervals were calculated using the recommended settings of GGDC 3.0 (22). Average nucleotide identity (ANI) values were calculated using FastANI (23). A core genome phylogeny of all strains was generated using M1CR0B1AL1Z3R (24). The parameters used included a maximum e-value cut-off of 0.01, a minimum percent identity of 70%, and a core gene presence threshold of 100% across all strains. The concatenated sequences of the resulting core proteomes were used to infer the phylogenetic relationships among the strains. Phylogenetic trees were constructed by concatenating the sequences of protein-coding genes shared across all strains. Tree inference was performed using the maximum-likelihood method, and bootstrap support values were calculated from 100 replicates to assess the robustness of the topology. Tree visualization and annotation were performed using the Interactive Tree of Life (iTOL) platform (25). A whole-genome phylogeny of representative strains was additionally assessed, using 107 single-copy core genes found in the majority of bacteria, using bcgTree (26). Visualization and annotation of the phylogenetic tree were performed using iTOL (25).

## Phenotypic characterization

The *A. schaalii*, *A. sanguinis*, and *A. urinale* type strains and representative isolates (i.e., the designated strains of genomic groups [see below]) of the seven novel *Actinotignum* species (see below) were cultivated on Columbia Chocolate agar medium and incubated at 37°C with 5% $CO_2$ for 48–72 h. Characteristics of the colonies were observed after incubation, and cell morphologies were determined by Gram staining. The metabolic profiles of the strains were assessed using the CCUG ANX3, COX, and LAX5 phenotypic worksheets (https://www.ccug.se/identification/worksheets), which include the API Rapid ID32AN, API CORYNE, and API Rapid ID32STREP commercial identification panels, following the protocols established by the manufacturer (bioMérieux, Marcy-l'Étoile, France). Antimicrobial susceptibility testing of 10 antimicrobials was performed, as described before (9).

## RESULTS

The analytical workflow consisted of a long-term collection of *Actinotignum* isolates, which were identified by 16S rRNA gene sequencing from 2010 to 2012 and by MALDI-TOF MS from 2013 to 2023. All collected isolates were then whole-genome sequenced and identified at the species level using a genome sequence-based internet tool (21). The taxonomic relationships between the isolates that could not be assigned to an established *Actinotignum* species were determined using dDDH (21) and ANI (22) analyses. These are well-established metrics in bacterial taxonomy that depict the degree of overall genomic relatedness between isolates, for which thresholds that can be applied generally are established below which isolates represent novel species. The main characteristics of the applied identification methods are depicted in Table 1.

A total of 137 *Actinotignum* isolates were collected and included in the present study. One hundred and sixteen (84.7%) and 21 (15.3%) were isolated from urine and non-urine samples, respectively. Non-urine samples included abdominal abscesses (*n* = 2), groin biopsies (*n* = 3), lower leg biopsies (*n* = 2), toe biopsies (*n* = 3), and one abdominal fascia, anal, anogenital, and perianal abscess, cavum uteri, ear canal, foot, jaw implant, mamma, tibia, and upper leg biopsy, each. Most samples (116, 84.7%) were from patients ≥60 years old.

## MALDI-TOF MS

Reanalysis of the MS spectra of 102 isolates using the version V12 database and version 4.1.100 MBT Compass software identified these isolates as *A. schaalii* (*n* = 95) or *A. sanguinis* (*n* = 7) with good (≥2) overall first identification scores (median values were 2.25 for *A. schaalii* and 2.14 for *A. sanguinis*). All individual identification reports included the following comment: "The species *A. sanguinis/schaalii* of the genus *Actinotignum* have very similar spectra. Therefore, discrimination on the species level is difficult."

## 16S rRNA gene sequence analyses

The 16S rRNA gene sequencing protocols allowed the determination of nearly complete sequences of 1439 to 1541 nucleotide positions for the proposed type strains (see below). These 16S rRNA gene sequences were 97.95%–99.11% identical to the 16S rRNA gene sequence of *A. schaalii* CCUG 27420[T] (CP008802), 98.08%–99.51% identical to the 16S rRNA gene sequence of *A. sanguinis* IMMIB L-2199[T] (HG798952); and 98.02%–

**TABLE 1** Overview of current identification methods used for the genus *Actinotignum*[b]

| Identification method | Availability | Speed | Cost | Resolution |
|---|---|---|---|---|
| MALDI-TOF MS | +++ | +++ | + | Genus[a] |
| 16S rRNA sequencing | ++ | ++ | ++ | Genus to species |
| Whole-genome sequencing | + | + | +++ | Species |

[a]Based on current database.
[b]+, low; ++, intermediate; +++, high.

99.93% identical to the 16S rRNA gene sequence of "*A. timonense*" Marseille-P2803[T] (LT706966) (Table S1). Only low values (93.42–94.30) were recorded toward the 16S rRNA gene sequence of *A. urinale* DSM 15805[T] (ATUY01000023). While the 16S rRNA gene sequences provided a reliable genus-level identification of the clinical isolates, they generally showed high percentages of sequence identity toward multiple type strains of *Actinotignum* species.

## Genomic analyses

The Illumina reads of the 137 *Actinotignum* isolates yielded draft genomes with 19–83 contigs and estimated genome sizes between 2.13 and 2.38 Mbp (Table S2). The percentage G + C content ranged from 60.93% to 62.30%. Except for two genomes with 2.89% and 1.77% contamination (isolate genomes usb_actsch_23 and usb_actsch_74, respectively), the contamination was consistently 0.5% or less, and the completeness varied between 98.63 and 99.94%.

Analysis of the 137 *Actinotignum* genomes through the TYGS server demonstrated that none of the isolates were identified as *A. schaalii* or *A. sanguinis*. In contrast, 82 (59.9%) were labeled as "*A. timonense*" and 55 (40.1%) as novel species. We calculated pairwise dDDH values among all 137 genomes and included the genome sequences of the type strain of *A. schaalii*, *A. sanguinis*, and *A. urinale*, as well as the "*A. timonense*" strain Marseille-P2803 genome (Table S3). Six groups of novel isolates comprising 82, 38, 8, 3, 3, and 2 isolates and a singleton (usb_actsch_93) were distinguished at the 70% dDDH threshold level for species delineation (27). dDDH values between isolates representing different groups were consistently below 70% (Table S3). Similarly, pairwise FastANI values were calculated between all genomes and revealed the same genomic groups of isolates and the singleton (Table S4), although FastANI values between isolates of some groups were slightly above the commonly applied species delineation threshold of 95%–96% (23, 28). FastANI calculations assume a probabilistic identity cutoff that is set to 80% by default, and, therefore, values below 80% are not shown in Table S4 (23). Table 2 shows the within-taxon and between-taxon ranges of FastANI and dDDH values toward each of the designated strains. Table S5 shows FastANI and dDDH values of public *Actinotignum* genome sequences toward the *A. schaalii*, *A. sanguinis*, and *A. urinale* type strains and the seven representative isolates of the present study, excluding repeat genome sequences of established *Actinotignum* type strains.

The isolates usb_actsch_203, usb_actsch_70, usb_actsch_31, usb_actsch_30, usb_actsch_61, and usb_actsch_34, along with the usb_actsch_93 singleton were chosen as representatives of the novel genomic groups. Table S6 shows the dDDH and FastANI values of all isolates with respect to these seven designated isolates, and toward the *A. schaalii*, *A. sanguinis*, and *A. urinale* type strains, as well as "*A. timonense*" strain Marseille-P2803. Figure 1 shows the result of the phylogenomic analysis based on 107 single-copy marker genes of the seven designated isolates, the *A. schaalii*, *A. sanguinis*, and *A. urinale* type strains and "*A. timonense*" strain Marseille-P2803. A core genome phylogeny analysis of all genomes confirmed the observed grouping of the

**TABLE 2** Within-taxon and between-taxon ranges of FastANI and dDDH values toward the representative strains of each of the novel *Actinotignum* species[b]

| Taxon representative (no. of isolates) | Proposed classification | FastANI (within-taxon range/between-taxon range) | dDDH (within-taxon range/between-taxon range) |
|---|---|---|---|
| usb_actsch_203 (82) | *A. lotii* | 99.17–97.20 (98.50)[a]/96.55–89.72 | 90.7–72.5 (84.3)[a]/67.8–22.3 |
| usb_actsch_70 (38) | *A. saccati* | 99.99–97.77 (98.63)/96.12–89.81 | 100–76.6 (85.9)/64.3–21.1 |
| usb_actsch_31 (8) | *A. inguinis* | 98.91–97.98 (98.59)/96.65–89.85 | 89.9–79.4 (85.7)/68.6–22.9 |
| usb_actsch_30 (3) | *A. vesiculae* | 99.85–97.30 (98.58)/95.91–90.26 | 98.3–72.7 (85.5)/61.2–21.8 |
| usb_actsch_61 (3) | *A. stranguriae* | 96.87–96.75 (96.82)/92.79–91.28 | 69.2–68.6 (68.9)/45.5–22.3 |
| usb_actsch_34 (2) | *A. cystesis* | 97.31/97.12–89.76 | 73.4/71.4–21.2 |
| usb_actsch_93 (1) | *A. urematis* | –/96.65–90.00 | –/67.5–21.0 |

[a]Average of within-taxon range.
[b]–, single-strain taxon.

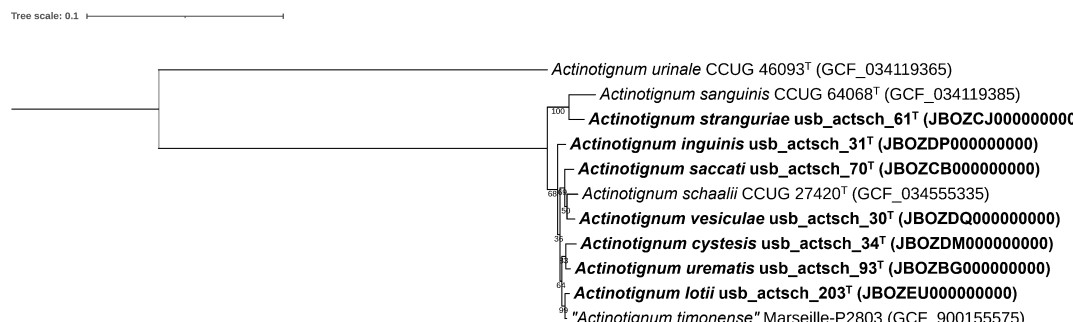

**FIG 1** *Actinotignum* phylogenomic tree. BcgTree (26) was used to extract the amino acid sequence of 107 single-copy core genes and perform a partitioned maximum-likelihood analysis on the concatenated amino acid sequences (35,990 positions). The phylogenetic tree was rooted at midpoint. The percentage of replicate trees in which the associated taxa clustered together in the bootstrap test (1,000 replicates) is shown next to the branches. Visualization and annotation of the tree were performed using iTOL (25). Genome accession numbers are given between parentheses. The scale bar indicates the number of substitutions per site.

137 new isolates and *Actinotignum* type and reference strains (Fig. S1). A total of 3,143 orthologous genes were detected, using a minimum identity cut-off of 70%. From these, 378 genes shared across all isolates were identified as core genes and concatenated into a data set comprising 136,425 amino acid residues. The resulting maximum-likelihood phylogenetic tree (Fig. S1) showed robust support across branches, with all bootstrap values exceeding 96% and most reaching 100%, highlighting the reliability of the inferred evolutionary relationships.

## Biochemical analysis

The 10 *Actinotignum* strains examined were gram-stain-positive rods with non-pigmented and non-hemolytic colonies. They were catalase-negative and grew best on Columbia blood agar at 37°C in a $CO_2$-enriched atmosphere or on Columbia Chocolate agar at 37°C in an anaerobic atmosphere. In the API ID32An microtest system, all strains exhibited activities of arginine and proline arylamidase, but none exhibited activities of L-arginine dihydrolase, α-galactosidase, β-galactosidase, β-galactosidase 6-phosphate, N-acetyl-β-glucosaminidase, D-mannose fermentation, nitrate reduction, L-tryptophanase (indole production), glutamic acid decarboxylase, or α-fucosidase. In the API Coryne microtest system, all strains fermented D-ribose, but none exhibited activities of catalase, nitrate reduction, β-galactosidase, N-acetyl-β-glucosaminidase, D-mannitol fermentation, D-lactose fermentation, or glycogen fermentation. Finally, in the API Rapid ID32Strep microtest system, all strains showed activity of hippuricase, but none showed activities of arginine arylamidase, β-glucosidase, α-galactosidase, β-galactosidase (both substrates), N-acetyl-β-glucosaminidase, β-mannosidase, fermentation of mannitol, sorbitol, lactose, raffinose, melibiose, melezitose, D-arabitol, tagatose, cyclodextrin, glycogen, pullulan or methyl-β-D-glucopyranoside, acetoin production, or glycyl tryptophan arylamidase. All other test results were variable between strains and are presented in Table S7.

## DISCUSSION

The present study set out to assess the performance of MALDI-TOF MS for the accurate species-level identification of novel *Actinotignum* isolates. To that end, we collected during a 14-year study period a total of 137 isolates from urine and other clinical specimens and identified these as *Actinotignum* bacteria using 16S rRNA sequence analysis (prior to 2012) or MALDI-TOF MS. We determined draft genome sequences of all 137 isolates and determined their species-level identities through whole-genome-based taxonomic comparisons using the TYGS classification server (21). Remarkably, none of the 137 isolates was identified as *A. schaalii*. In contrast, 82 (59.9%) isolates

were identified as "*A. timonense*," and 55 (40.1%) isolates represented novel *Actinotignum* species.

Pairwise comparisons of dDDH and FastANI values between all genomes revealed that the 82 isolates and "*A. timonense*" strain Marseille-P2803 shared dDDH values of at least 72.5% with strain usb_actsch_203, a designated representative of this group (Table 2; Table S3), indicating that they represent a single species (27, 29). These isolates shared FastANI values of at least 97.20% toward the same representative strain (Table 2; Table S4). Similarly, isolates of the genomic groups represented by strains usb_actsch_70 (including 38 isolates), usb_actsch_31 (including 8 isolates), and usb_actsch_30 (including 3 isolates) shared dDDH values of at least 76.6%, 79.4%, and 72.7%, respectively (Table 2; Table S3); while they shared FastANI values of at least 97.77%, 97.98%, and 97.30% toward the respective representative strains (Table 2; Table S4). Isolates of the genomic group represented by strain usb_actsch_61 (including three isolates) were somewhat aberrant in that they shared dDDH values of at least 68.6%, while they shared FastANI values of at least 96.76% with the designated representative strain (Table 2; Tables S3 and S4). dDDH values between isolates of different genomic groups with at least three isolates, the singleton (usb_actsch_93) and the group represented by strain usb_actsch_34, were (except for a single value of 71.4% [Table S3]) consistently below 70% and, thus, confirmed that these seven genomic groups are best classified as seven novel *Actinotignum* species. While there were clear gaps between the within-taxon and between-taxa FastANI values (Table 2; Table S4), many between-taxa FastANI values were slightly above the generally applied threshold of 95%–96% (23, 28). Similar slight discrepancies between species delineation thresholds based on average nucleotide identity and dDDH overall genome relatedness indices have been reported in other groups of bacteria (30) and highlight that species delineation thresholds should not be applied blindly. Below, we propose the names *A. cystesis* sp. nov., *A. inguinis* sp. nov., *A. saccati* sp. nov., *A. stranguriae* sp. nov., *A. urematis* sp. nov., and *A. vesiculae* sp. nov., for the species represented by strains usb_actsch_34[T], usb_actsch_31[T], usb_actsch_70[T], usb_actsch_61[T], usb_actsch_93[T], and usb_actsch_30[T], respectively. We propose the novel species name, *A. lotii* sp. nov., for the group of 82 novel isolates represented by isolate usb_actsch_203[T] and "*A. timonense*" strain Marseille-P2803. We prefer not to use the name "*A. timonense*" (5) for this novel species, as a temporary Expression of Concern was published in March 2024 (31) regarding this study.

A repeat analysis of the mass spectra of 102 isolates that remained preserved, using the version V12 database and the version 4.1.100 MBT Compass software, identified all isolates as *A. schaalii* or *A. sanguinis* with good (≥2) overall first identification scores (median values were 2.25 for *A. schaalii* and 2.14 for *A. sanguinis*). These results, therefore, confirmed that the MALDI-TOF MS-based species-level identification of bacteria of the *Actinotignum* genus is unreliable. Commercial databases aiming at species-level identifications of human microbiome bacteria should be extended with reference spectra of all *Actinotignum* species, and the identities of the reference strains and confirmed genomic species used for construction of the present commercial databases should be reassessed using whole-genome sequence-based criteria. At present, it is unclear whether such an extension and correction of the identification database will facilitate accurate species-level identification.

The extensive phenotypic analysis of the type strains of the proposed novel species revealed minimal differential biochemical characteristics (Table S7). Although whole-genome sequence analysis is still beyond the capabilities of many routine clinical laboratories, we believe that it is currently the only reliable method of species identification within this genus and can now be used to assess the functional or pathogenic potential of each of these species. Alternatively, future studies can exploit the present taxonomic framework and genomic data to try and develop a simple and inexpensive PCR system and Sanger sequencing to provide species identification within the genus *Actinotignum*. A reanalysis of public *Actinotignum* genome sequences, using the reference genomes and species delineation thresholds determined in the present study,

identified 19 of 52 genomes of clinical isolates as *A. lotii*, including two that are listed as *A. schaalii* in NCBI (Table S5). The identifications of six isolates labeled as *A. urinale* and of 15 isolates as *A. sanguinis* were confirmed, but for 11 isolates labeled as *A. schaalii* (5 isolates) or "*A. timonense*" (6 isolates), the identification was incorrect (Table S5). Finally, a single isolate currently labeled as *Actinotignum* sp. and three isolates labeled as *A. schaalii* represent additional novel *Actinotignum* species (Table S5).

To conclude, the present study revealed limitations of MALDI-TOF MS for species-level identification of *Actinotignum* using current databases and highlighted an impressive genomic diversity among human clinical *Actinotignum* isolates. In addition, we did not identify *A. schaalii* or *A. sanguinis* infections in the studied patient cohort. Our study did not allow for a retrospective evaluation of genetic or clinical characteristics of the newly defined *Actinotignum* species. However, there have been reports on genetic and clinical diversity between strains identified as *A. schaalii* (3, 32), and the present MIC analysis of newly proposed type strains revealed elevated MICs for trimethoprim-sulfamethoxazole, ciprofloxacin, and clindamycin in *A. cystesis* usb_actsch_34$^T$, for clindamycin in *A. saccati* usb_actsch_70$^T$, and for trimethoprim-sulfamethoxazole and ciprofloxacin in the type strains of another four novel *Actinotignum* species (Table S7).

By providing an extended taxonomic framework for the *Actinotignum* genus, the present study may allow the assignment of such genetic or clinical diversity to described *Actinotignum* species. Future studies will need to assess if better defined species-specific treatment guidelines can be introduced so that ultimately patient morbidity can be reduced or ineffective antibiotic usage can be prevented.

## Protologs of novel *Actinotignum* species

The phenotypic characteristics of the type strains are as described above and in Table S7.

## Description of *Actinotignum cystesis* sp. nov

*Actinotignum cystesis* (cys.te'sis. L. gen. n. *cystesis*, from a bladder).

The type strain is usb_actsch_34$^T$ (LMG 33441$^T$ = CCUG 77579$^T$) isolated from urine of a 60–79-year-old human in Basel, Switzerland, in 2017. A second isolate was also obtained from a urine sample. The 16S rRNA gene and whole-genome sequence of strain usb_actsch_34$^T$ are publicly available through the accession numbers PV889331 and JBOZDM000000000, respectively. The G + C content of the DNA of the type strain is 62.06 mol%. Its genome size is 2.17 Mbp.

## Description of *Actinotignum inguinis* sp. nov

*Actinotignum inguinis* (in.gui'nis. L. gen. n. *inguinis* from a groin).

The type strain usb_actsch_31$^T$ (LMG 33440$^T$ = CCUG 77578$^T$) was isolated from the groin of a 40–59-year-old human in Basel, Switzerland, in 2017. Other isolates were obtained from urine samples. The 16S rRNA gene and whole-genome sequence of strain usb_actsch_31$^T$ are publicly available through the accession numbers PV889330 and JBOZDP000000000, respectively. The G + C content of the DNA of the type strain is 62.12 mol%. Its genome size is 2.18 Mbp.

## Description of *Actinotignum lotii* sp. nov

*Actinotignum lotii* (lo'ti.i. L. gen. n. *lotii* from urine).

The type strain usb_actsch_203$^T$ (LMG 33445$^T$ = CCUG 77583$^T$) was isolated from urine of a 60–79-year-old human in Basel, Switzerland, in 2023. Other isolates were obtained from urine, anal and perianal abscesses, an anogenital biopsy, cavum uteri and groin swabs, an ear canal swab, a jaw implant, an abdominal fascia aspirate, and upper and lower leg and toe wounds. The 16S rRNA gene and whole-genome sequence of strain usb_actsch_203$^T$ are publicly available through the accession numbers PV889335 and JBOZEU000000000, respectively. The G + C content of the DNA of the type strain is 61.67 mol%. Its genome size is 2.34 Mbp.

## Description of *Actinotignum saccati* sp. nov

*Actinotignum saccati* (sac.ca'ti. L. gen. n. *saccati* from urine).

The type strain usb_actsch_70<sup>T</sup> (LMG 33443<sup>T</sup> = CCUG 77581<sup>T</sup>) was isolated from urine of a 60–79-year-old human in Basel, Switzerland, in 2017. Other isolates were obtained from urine, abdominal wall abscesses, lower leg, foot and toe wounds, and breast and groin wounds. The 16S rRNA gene and whole-genome sequence of strain usb_actsch_70<sup>T</sup> are publicly available through the accession numbers PV889333 and JBOZCB000000000, respectively. The G + C content of the DNA of the type strain is 61.96 mol%. Its genome size is 2.31 Mbp.

## Description of *Actinotignum stranguriae* sp. nov

*Actinotignum stranguriae* (stran.gu'ri.ae. L. gen. n. *stranguriae* from a disease of urinary organs).

The type strain usb_actsch_61<sup>T</sup> (LMG 33442<sup>T</sup> = CCUG 77580<sup>T</sup>) was isolated from urine of an 80–99-year-old human in Basel, Switzerland, in 2017. Other isolates were also obtained from urine samples. The 16S rRNA gene and whole-genome sequence of strain usb_actsch_61<sup>T</sup> are publicly available through the accession numbers PV889332 and JBOZCJ000000000, respectively. The G + C content of the DNA of the type strain is 61.08 mol%. Its genome size is 2.20 Mbp.

## Description of *Actinotignum urematis* sp. nov

*Actinotignum urematis* (u.re'ma.tis. Gr. gen. n. *urematis* from urine).

The type strain usb_actsch_93<sup>T</sup> (LMG 33444<sup>T</sup> = CCUG 77582<sup>T</sup>) was isolated from urine of an 80–99-year-old human in Basel, Switzerland, in 2018. The 16S rRNA gene and whole-genome sequence of strain usb_actsch_93<sup>T</sup> are publicly available through the accession numbers PV889334 and JBOZBG000000000, respectively. The G + C content of the DNA of the type strain is 62.08 mol%. Its genome size is 2.18 Mbp.

## Description of *Actinotignum vesiculae* sp. nov

*Actinotignum vesiculae* (ve.si'cu.lae. L. gen . n. *vesiculae* from a bladder).

The type strain usb_actsch_30<sup>T</sup> (LMG 33439<sup>T</sup> = CCUG 77577<sup>T</sup>) was isolated from urine of a 60–79-year-old human in Basel, Switzerland, in 2017. Other isolates were also obtained from urine samples. The 16S rRNA gene and whole-genome sequence of strain usb_actsch_30<sup>T</sup> are publicly available through the accession numbers PV889329 and JBOZDQ000000000, respectively. The G + C content of the DNA of the type strain is 62.17 mol%. Its genome size is 2.16 Mbp.

## ACKNOWLEDGMENTS

We thank the entire team from the Clinical Bacteriology/Mycology Unit from the University Hospital Basel for excellent technical assistance and scientific support. Bioinformatic analyses were partially performed at sciCORE (http://scicore.unibas.ch), the scientific computing center at the University of Basel. Other computational resources and services used in this study were provided by the VSC (Flemish Supercomputer Center), funded by the Research Foundation Flanders (FWO) and the Flemish Government—department EWI. The authors acknowledge the staff at the Culture Collection University of Gothenburg (CCUG) Lab for maintaining and phenotyping the strains used in this study.

G.S-C. and E.R.B.M. were supported by funding under the agreement between the Swedish government and the Avtal om Läkarutbildning och Forskning (ALF; ALFGBG-966570) and Forskning och Utveckling (FoU; VGRFOUREG-1013833) of Västra Götalandsregionen (VGR). The Culture Collection University of Gothenburg (CCUG; https://www.ccug.se/) is supported by the Department of Clinical Microbiology, Sahlgrenska University Hospital and the Sahlgrenska Academy of the University of

Gothenburg; G.S-C. was supported by the Genomics & Proteomics Research Project of the Bacterial Diversity Program of the CCUG.

D.G.: conceptualization study, supervision, writing original draft. D.M.W.: data collection and analysis, writing original draft. P.V.: genome data analysis, writing original draft. E.R.B.M.: phenotypic analyses. D.G., C.L., B.I.: microbiological data collection and analysis, MIC analysis. D.M.W., G.S-C., C.P., P.M.K., P.S.: MALDI-TOF MS and genome data collection and analysis. All authors reviewed the final draft.

## AUTHOR AFFILIATIONS

[1]Clinical Bacteriology and Mycology, University Hospital Basel and University of Basel, Basel, Switzerland

[2]Department of Clinical Microbiology, Culture Collection University of Gothenburg (CCUG), Sahlgrenska University Hospital, Region Västra Götaland, Gothenburg, Sweden

[3]Department of Infectious Diseases, Institute for Biomedicine, Sahlgrenska Academy, University of Gothenburg, Gothenburg, Sweden

[4]Laboratory of Microbiology, Department of Biochemistry and Microbiology, Ghent University, Ghent, Belgium

[5]Laboratory Medicine, University Hospital Basel and University of Basel, Basel, Switzerland

## AUTHOR ORCIDs

Dylan M. Winterflood (ID) http://orcid.org/0009-0002-1392-1311
Charlotte Peeters (ID) http://orcid.org/0000-0002-1891-4869
Peter M. Keller (ID) http://orcid.org/0000-0002-2890-5384
Pascal Schlaepfer (ID) http://orcid.org/0000-0002-0828-8681
Edward R. B. Moore (ID) https://orcid.org/0000-0001-7693-924X
Daniel Goldenberger (ID) http://orcid.org/0000-0002-3788-3818

## AUTHOR CONTRIBUTIONS

Dylan M. Winterflood, Data curation, Formal analysis, Methodology, Software, Validation, Writing – original draft | Guillem Seguí-Crespi, Data curation, Formal analysis, Software, Validation, Writing – review and editing | Charlotte Peeters, Data curation, Software, Validation, Writing – review and editing | Claudia Lang, Data curation, Formal analysis, Methodology, Validation, Writing – review and editing | Branislav Ivan, Data curation, Formal analysis, Methodology, Writing – review and editing | Peter M. Keller, Data curation, Formal analysis, Validation, Writing – review and editing | Pascal Schlaepfer, Data curation, Formal analysis, Investigation, Software, Validation, Writing – review and editing | Edward R. B. Moore, Data curation, Formal analysis, Investigation, Validation, Writing – review and editing | Peter Vandamme, Data curation, Formal analysis, Investigation, Methodology, Supervision, Validation, Writing – original draft, Writing – review and editing | Daniel Goldenberger, Conceptualization, Data curation, Formal analysis, Methodology, Supervision, Writing – original draft

## DATA AVAILABILITY

All 137 genome sequences determined in the present study are available at the European Nucleotide Archive (ENA) under the BioProject PRJNA1162065. 16S rRNA gene sequences determined in the present study are available under the accession numbers PV889329–PV889335.

## ETHICS APPROVAL

The study was approved by the local ethics committee named Ethikkommission Nordwest- und Zentralschweiz (EKNZ), BASEC Project ID no. Req-2024-00492, according to the standards of the Swiss Human Research Act.

## ADDITIONAL FILES

The following material is available online.

### Supplemental Material

**Figure S1 (Spectrum02190-25-s0001.pdf).** Core genome phylogeny.
**Supplemental legend (Spectrum02190-25-s0002.docx).** Descriptive legend for Fig. S1.
**Supplemental tables (Spectrum02190-25-s0003.xlsx).** Tables S1 to S7.

### Open Peer Review

**PEER REVIEW HISTORY (review-history.pdf).** An accounting of the reviewer comments and feedback.

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
