## [Reviewer comments · Microbiology Spectrum]

Microbiology Spectrum

Whole-genome sequence analyses reveal cryptic species-level diversity among clinical *Actinotignum* isolates misidentified by matrix-assisted laser desorption/ionization time-of-flight mass spectrometry

Dylan Winterflood, Guillem Segui-Crespi, Charlotte Peeters, Claudia Lang, Branislav Ivan, Peter Keller, Pascal Schläpfer, Edward Moore, Peter VANDAMME, and Daniel Goldenberger

Corresponding Author(s): Daniel Goldenberger, Universitatsspital Basel Abteilung Klinische Mikrobiologie

Review Timeline:

Submission Date:	July 17, 2025
Editorial Decision:	December 5, 2025
Revision Received:	December 13, 2025
Editorial Decision:	December 26, 2025
Revision Received:	January 8, 2026
Accepted:	January 23, 2026

Editor: Ayesha Khan

Reviewer(s): Disclosure of reviewer identity is with reference to reviewer comments included in decision letter(s). The following individuals involved in review of your submission have agreed to reveal their identity: Nidhika Berry (Reviewer #2)

Transaction Report:

DOI: <https://doi.org/10.1128/spectrum.02190-25>

Re: Spectrum02190-25 (**Whole-genome sequence but not matrix-assisted laser desorption/ionization time-of-flight mass spectrometry-based analysis of 137 *Actinotignum* isolates from human clinical samples identifies seven novel *Actinotignum* species**)

Dear Dr. Daniel Goldenberger:

Thank you for the privilege of reviewing your work. Below you will find my comments, instructions from the Spectrum editorial office, and the reviewer comments.

Revision Guidelines

Sincerely,
Ayesha Khan
Editor
Microbiology Spectrum

Reviewer #1 (Comments for the Author):

This is a well-written manuscript describing a well-executed study of a poorly characterized genus of organisms commonly associated with urinary tract infections in the elderly.

The authors should, however, consider minor modifications to the text, or ideally a not-so-minor addition to the study, as follows:

(line 60) "Currently, whole-genome sequence based identification may need to be used to provide correct species identifications and to assess the genetic and clinical differences between established and newly defined *Actinotignum* species."

Similarly, (line 337) "we believe whole-genome sequence analyses should form the basis for reliable species identifications within this genus and be used to assess the functional or pathogenic potential of each of these species."

While it may be true that WGS analysis is needed to assess genetic, clinical, functional, or pathogenic potential, it is likely that WGS analysis could be used to identify 1 or 2 core loci that, following simple and inexpensive PCR amplification and Sanger sequencing, would provide species identification without the complexity and cost of WGS. Furthermore, this alternative approach would be amenable to culture-independent identification of these difficult to culture bacteria.

Reviewer #2 (Comments for the Author):

In the methodology, were these urine specimens cultured specifically for this purpose/ study? Usually we do not (as noted by the authors in introduction) isolate these isolates from chromogenic agar and CLED plates. Was there a previous study designed for urine isolates to be gathered using anaerobic cultures or CO₂ and enrichment of urine as done to pick up the *Actinotignum* spp? Was there a clinical reason eg recurrent UTI, non response, or other reason to specifically do this in addition or the laboratory protocol includes this for all specimens?

Are urine samples routinely cultured aerobically and anaerobically in your laboratories?

Would there be any bacteraemia isolates as well, which could be studied?

The authors have mentioned the clinical data is not available, but some MIC data is available in the text. Any susceptibility data and treatment response data for each of the novel species would be quite useful for the clinical microbiologists, to ascertain empiric treatment, as WGS is not readily available in clinical laboratories and results take some time.

Point-by-point responses to manuscript Spectrum02190-25, Whole-genome sequence but not matrix-assisted laser desorption/ionization time-of-flight mass spectrometry-based analysis of 137 *Actinotignum* isolates from human clinical samples identifies seven novel *Actinotignum* species

Dear Dr. Ayesha Khan

Thank you for the time to edit and review our manuscript. We appreciate the comments and believe they will improve the quality of the manuscript. Here, we provide point-by-point responses to the comments from your reviewers.

Reviewer #1 (Comments for the Author):

This is a well-written manuscript describing a well-executed study of a poorly characterized genus of organisms commonly associated with urinary tract infections in the elderly.

The authors should, however, consider minor modifications to the text, or ideally a not-so-minor addition to the study, as follows:

(line 60) "Currently, whole-genome sequence based identification may need to be used to provide correct species identifications and to assess the genetic and clinical differences between established and newly defined *Actinotignum* species."

Similarly, (line 337) "we believe whole-genome sequence analyses should form the basis for reliable species identifications within this genus and be used to assess the functional or pathogenic potential of each of these species."

While it may be true that WGS analysis is needed to assess genetic, clinical, functional, or pathogenic potential, it is likely that WGS analysis could be used to identify 1 or 2 core loci that, following simple and inexpensive PCR amplification and Sanger sequencing, would provide species identification without the complexity and cost of WGS. Furthermore, this alternative approach would be amenable to culture-independent identification of these difficult to culture bacteria.

Response:

*Thank you for this important comment. Our study is focused on the performance of MALDI-TOF MS currently recommended for routine identification in comparison to the gold standard WGS for species identification within the genus *Actinotignum*. Our work showed for the first time that WGS represent the method of choice for species identification of this genus and provided data to describe 7 novel *Actinotignum* species. We believe that your input would be a valuable contribution in a future study using our novel genome data for precise and rapid identification of *Actinotignum* spp. We therefore added the following text to our manuscript from line 348:*

In a future study, our novel genome data might be exploited to develop a simple and inexpensive PCR system and Sanger sequencing to provide species identification

within the genus Actinotignum without the complexity and cost of WGS analysis. Furthermore, this alternative method would be amenable to culture-independent detection of these difficult to culture bacteria.

Reviewer #2 (Comments for the Author):

In the methodology, were these urine specimens cultured specifically for this purpose/ study?

Response:

These urine samples have been cultured during routine work and were then collected and refrigerated for the present study.

Usually we do not (as noted by the authors in introduction) isolate these isolates from chromogenic agar and CLED plates. Was there a previous study designed for urine isolates to be gathered using anaerobic cultures or CO₂ and enrichment of urine as done to pick up the Actinotignum spp?

Response:

Thank you for this important comment. We perform urine culture routinely using sheep blood agar (in 5% CO₂ atmosphere) and chromogenic agar at 37°C. Therefore, we were able to culture this high amount of Actinotignum isolates from urine samples. Based on your comment and to complete our manuscript within the materials and methods part, we added the following text from line 100:

Urine samples were cultured on Columbia sheep blood agar, incubated in 5% CO₂ atmosphere and CHROMagar™ Orientation medium according to standard methods.

Was there a clinical reason eg recurrent UTI, non response, or other reason to specifically do this in addition or the laboratory protocol includes this for all specimens?

Response:

Our laboratory protocol includes this for all urine samples (see text above).

Are urine samples routinely cultured aerobically and anaerobically in your laboratories?

Response:

All inoculated sheep blood agar plates are routinely incubated at 5% CO₂ and inoculated CHROMagar Orientation agar plates are routinely incubated aerobically.

Would there be any bacteraemia isolates as well, which could be studied?

Response:

In our study, we could not culture isolates from the blood and therefore were not able to include such strains to our present work. But we agree, It would be interesting to investigate multiple blood culture isolates in comparison to non-blood culture strains using WGS in a next study.

The authors have mentioned the clinical data is not available, but some MIC data is available in the text. Any susceptibility data and treatment response data for each of the novel species would be quite useful for the clinical microbiologists, to ascertain empiric treatment, as WGS is not readily available in clinical laboratories and results take some time.

Response:

We present in Supplemental Table S7 detailed antimicrobial susceptibility data from the 7 novel Actinotignum spp. type strains as well as from the two type strains representing A. schaalii and A. sanguinis. A total of 10 antimicrobials have been determined. Analyzing the resulting minimum inhibitory concentrations (MICs), we confirm from these novel species previous data from Lotte R. et al. (reference no. 4 in the manuscript) published as review article. All beta lactams show low MICs, but both trimethoprim-sulfamethoxazol and ciprofloxacin show in 5 from 7 novel isolates MICs >32mg/l. Due to the focus of our study on microbiology and taxonomy of Actinotignum, we are not able to provide clinical data such as treatment response data. But we agree, a thorough clinical investigation on symptoms, epidemiology, therapy and outcome using our novel microbiological data would be very welcome.

Re: Spectrum02190-25R1 (**Whole-genome sequence but not matrix-assisted laser desorption/ionization time-of-flight mass spectrometry-based analysis of 137 *Actinotignum* isolates from human clinical samples identifies seven novel *Actinotignum* species**)

Dear Dr. Daniel Goldenberger:

Thank you for the privilege of reviewing your work. Below you will find my comments which should be addressed prior to final acceptance, and instructions from the Spectrum editorial office.

Editor comments:

1. I suggest making minor modifications to the discussion section. Start by emphasizing conclusions drawn directly by your data in this study. I.e. limitations of MALDI-TOF MS for species-level identification of *Actinotignum* using current databases, and the diversity revealed by WGS of clinical isolates. If discussing other points like pathogenic potential, clinical relevance, or future treatment implications- clearly frame these as hypotheses, future directions, etc.
2. The manuscript emphasizes that WGS should be standard for *Actinotignum* species identification. While the data is compelling, I recommend softening language around recommending WGS as a method of choice. Explicitly acknowledge the practical limitations of WGS in routine clinical laboratories. This could be added to the section where future PCR/Sanger-based alternatives are mentioned.
3. Given the journal's diverse readership, consider clarifying the following:
 - Add a brief summary sentence at the start of the Results section explaining the analytical workflow from MALDI-TOF to WGS to genomic species identification.
 - Where ANI and dDDH thresholds are discussed, ensure that their interpretive significance is clearly explained for readers less familiar with these concepts. Explain why their use together is important for genomic taxonomy.
4. Ensure consistent use of species names and abbreviations throughout (especially when referring to "*A. timonense*" versus *A. lotii*).
5. Please double-check that all supplementary tables and figures are clearly referenced at first mention.
6. Minor suggestion: Consider adding a single schematic figure or table-formal summary breaking down how MALDI, 16S rRNA sequencing, and WGS differ in their resolving power for *Actinotignum* species (benefits, limitations).
7. Please clarify in discussion if failure of MALDI reflects database limitations rather than intrinsic flaws of the technology itself.
8. I suggest modifying the title such that the main takeaway of the manuscript is clear. For example: "Whole-genome sequencing reveals cryptic species-level diversity among clinical *Actinotignum* isolates misidentified by MALDI-TOF MS"

Revision Guidelines

For complete guidelines on revision requirements, see our Submission and Review Process webpage. Submission of a paper

that does not conform to guidelines may delay acceptance of your manuscript.

Sincerely,
Ayesha Khan
Editor
Microbiology Spectrum

Point-by-point responses to manuscript Spectrum02190-25

Dear Dr. Ayesha Khan

Thank you for your review and your valuable comments. Below please find the point-by-point responses to your editorial comments

1. I suggest making minor modifications to the discussion section. Start by emphasizing conclusions drawn directly by your data in this study. I.e. limitations of MALDI-TOF MS for species-level identification of *Actinotignum* using current databases, and the diversity revealed by WGS of clinical isolates. If discussing other points like pathogenic potential, clinical relevance, or future treatment implications- clearly frame these as hypotheses, future directions, etc.

Author's response:

We modified the concluding paragraph of the discussion section to explicitly distinguish findings of the present study from future research directions and potential treatment implications.

2. The manuscript emphasizes that WGS should be standard for *Actinotignum* species identification. While the data is compelling, I recommend softening language around recommending WGS as a method of choice. Explicitly acknowledge the practical limitations of WGS in routine clinical laboratories. This could be added to the section where future PCR/Sanger-based alternatives are mentioned.

Author's response:

We revised the text to soften language around recommending WGS as a method of choice and acknowledged the limitations of WGS in many routine clinical laboratories today.

3. Given the journal's diverse readership, consider clarifying the following:

- Add a brief summary sentence at the start of the Results section explaining the analytical workflow from MALDI-TOF to WGS to genomic species identification.
- Where ANI and dDDH thresholds are discussed, ensure that their interpretive significance is clearly explained for readers less familiar with these concepts. Explain why their use together is important for genomic taxonomy.

Author's response:

A brief summary of the analytical workflow was included at the start of the Results section. This novel text paragraph also includes some basic background information on the ANI and dDDH thresholds.

4. Ensure consistent use of species names and abbreviations throughout (especially when referring to "A. timonense" versus A. lotii).

Author's response:

We double-checked the manuscript and consistently abbreviated the genus name '*Actinotignum*' to 'A.'.

5. Please double-check that all supplementary tables and figures are clearly referenced at first mention.

Author's response:

We double-checked the text according to your comment and confirm correct references of supplementary data concerning order and designation. The term Supplementary was deleted, when used in second or third mention.

6. Minor suggestion: Consider adding a single schematic figure or table-formal summary breaking down how MALDI, 16S rRNA sequencing, and WGS differ in their resolving power for *Actinotignum* species (benefits, limitations).

Author's response:

We added this table to the text.

7. Please clarify in discussion if failure of MALDI reflects database limitations rather than intrinsic flaws of the technology itself.

Author's response:

We actually do not have the data to provide an answer to this question today, and clarified this in the revised manuscript.

8. I suggest modifying the title such that the main takeaway of the manuscript is clear. For example: "Whole-genome sequencing reveals cryptic species-level diversity among clinical *Actinotignum* isolates misidentified by MALDI-TOF MS"

Author's response:

We modified the title as suggested.

Re: Spectrum02190-25R2 (**Whole-genome sequence analyses reveal cryptic species-level diversity among clinical *Actinotignum* isolates misidentified by matrix-assisted laser desorption/ionization time-of-flight mass spectrometry**)

Dear Dr. Daniel Goldenberger:

Your manuscript has been accepted, and I am forwarding it to the ASM production staff for publication. Your paper will first be checked to make sure all elements meet the technical requirements. ASM staff will contact you if anything needs to be revised before copyediting and production can begin. Otherwise, you will be notified when your proofs are ready to be viewed.

Sincerely,
Ayesha Khan
Editor
Microbiology Spectrum